# Old Ways, New Ways—Scaling Up from Customary Use of Plant Products to Commercial Harvest Taking a Multifunctional, Landscape Approach

**Julian Gorman** [1,2,*], **Diane Pearson** [3] **and Penelope Wurm** [2]

1 School of People Environment and Planning, Massey University, Palmerston North 4442, New Zealand
2 Research Institute for the Environment and Livelihoods, Charles Darwin University, Darwin NT 0810, Australia; Penny.Wurm@cdu.edu.au
3 Farmed Landscape Research Centre, School of Agriculture and Environment, Massey University, Palmerston North 4442, New Zealand; d.pearson@massey.ac.nz
* Correspondence: julian.gorman@cdu.edu.au; Tel.: +64-212571719

**Abstract:** Globally, the agricultural sector is facing many challenges in response to climate change, unsustainable farming practices and human population growth. Despite advances in technology and innovation in agriculture, governments around the world are recognizing a need for transformative agricultural systems that offer solutions to the interrelated issues of food security, climate change, and conservation of environmental and cultural values. Approaches to production are needed that are holistic and multisectoral. In planning for future agricultural models, it is worth exploring indigenous agricultural heritage systems that have demonstrated success in community food security without major environmental impacts. We demonstrate how indigenous practices of customary harvest, operating in multifunctional landscapes, can be scaled up to service new markets while still maintaining natural and cultural values. We do this through a case analysis of the wild harvest of Kakadu plum fruit by Aboriginal people across the tropical savannas of northern Australia. We conclude that this system would ideally operate at a landscape scale to ensure sustainability of harvest, maintenance of important patterns and processes for landscape health, and incorporate cultural and livelihood objectives. Applied to a variety of similar native products, such a production system has potential to make a substantial contribution to niche areas of global food and livelihood security.

**Keywords:** agricultural systems; indigenous economic development; production systems; landscape ecology; wild harvest

## 1. Introduction

Population growth, climate change, and unsustainable farming practices are some of the most pressing global challenges in the 21st century. Governments from around the globe have acknowledged these 'wicked' problems and are developing strategies to mitigate their impacts [1–3]. The agricultural sector is one area that is under considerable pressure to adjust and perform to meet the increased challenges of sustainable food production practices under increasingly uncertain climatic conditions. However, food production is currently occurring in a way which is having adverse impacts on climate, water, topsoil, biodiversity and marine environments [4]. If not addressed, these practices will undermine the world's ability to adequately feed future populations, and solutions are desperately needed to provide more sustainable options.

*1.1. Food Security*

The global population is predicted to reach 9.73 billion people by 2050 and will require more food, more stock feed and more biofuel to meet these demands [1]. Globally, food security is currently dependent on a small number of cultivated species with only 12 species contributing to 80% of total dietary intake [5]. Wheat, maize and rice account for over 50% of the world's daily requirement of protein and calories [6]. Despite there being limitations in crop–climate modeling, the impacts of climate change are almost certain to decrease global crop production [7]. This is a real concern for food security with a growing global population and there is now substantial research into global plant genetic resources being conducted, focused on improved cultivars, breeding lines, landraces and crop wild relatives diversity [8–10].

Global per capita consumption has increased considerably since the 1970s, with doubling of milk, dairy products and vegetables, while meats products have tripled (Alexandratos and Bruinsma in [1]). However, total productivity resulting from agricultural investment and technology is now thought to have slowed as a result of food loss and wastage, degradation of natural resources, biodiversity loss and spread of transboundary pests and disease, and resistance to antimicrobials [1].

There is now a marked global disparity of agricultural outputs among countries. Developed countries are currently facing chronic obesity rates, which have more than doubled since 1980s [11]. Issues added to these human health problems include an increased carbon footprint and an increase in area lost to landfill and biodiversity losses. Some 25–30% of food produced worldwide goes to waste, which costs about US$1 trillion per year and accounts for 10% of the greenhouse gas emission from food systems [12]. Third world countries continue to suffer from poverty and food shortages as well as chronic undernutrition [4]. Given the serious threat to food security and the far-reaching impacts of climate change on crops, livestock and fisheries production, agriculture needs to balance research and implementation strategies to be able to face these challenges [2].

*1.2. Impacts of Climate Change*

The impacts of climate change on agriculture are predicted to be significant. Climate change is a certainty [12] and the agriculture sector is, and will continue to be, impacted in various ways. The Intergovernmental Panel on Climate Change (IPCC) predicts a temperature rise of +1.5 °C above pre-industrial levels sometime between 2030 and 2052, if global warming continues at its current rate. The impacts of such changes, despite increasing confidence in prediction modeling, are not altogether certain. In 2019, the IPCC Special Report provided new evidence for the benefits of limiting global warming to the lowest possible level, in line with the goals set in the 2015 Paris Agreement [12,13].

There are numerous predictions of climate change that will impact on the agricultural sector. These include more extreme weather events with both drought and floods becoming more common in areas making it more expensive and difficult to grow and sustain crops and livestock. A change in weather will influence growing seasons and cause an impact upon productivity due to non-alignment of crop growth with soil moisture levels and pests. In some areas, seasonal weather patterns may cause an increase in the frequency of wildfires that will result in physical damage to infrastructure and pasture as well as a number of 'secondary impacts' from smoke [6]. Rising temperatures can alter exposure to pathogens and toxins, and rising levels of carbon dioxide in the atmosphere can decrease dietary iron, zinc, protein and other macro- and micro-nutrients [1,14].

*1.3. Environmental Impacts*

The rapid global expansion in food production and economic growth that the world has seen since the 1960s has come at a heavy cost to the natural environment. Adoption of high input and resource intense farming systems have caused massive deforestation, soil depletion, water scarcities and contributed to high levels of greenhouse gas emissions [1]. Klitgaard [15] suggests that systems in overshoot such as these require new economic theories to achieve sustainable futures.

As agriculture has expanded in recent decades, there has been greater competition for natural resources, increased carbon emissions, and land degradation. There has been a narrowing of cropping choice [9] and lack of diversity in crop rotation, which is coincident with an overuse of pesticides and other chemicals, which is damaging to human and ecosystem health [16].

Globally, agriculture is responsible for using 70% of all freshwater withdrawn from the natural system and 60% of biodiversity loss [4]. Half of the world's forests have been cleared, an ever-increasing volume of greenhouse gases is going into the atmosphere and ground water has been contaminated or depleted. However, agricultural land use has resulted in land values increasing as the system has become more capital-intensive, and requires greater vertical integration, which leads to big cooperatives dictating land-use. This is impacting on the social and cultural structure of rural towns, removing safety nets and increasing levels of rural poverty. This then drives migration into large cities which exacerbates welfare, food, employment and health issues.

Agricultural expansion is resulting in habitat loss [17–19], which in turn, has been identified as the primary contributor to what has been described as the 'Anthropocene' [20] or sixth mass extinction event [21].

The Australian agricultural and rural sectors are following the global trends outlined above and are currently facing extreme social and economic pressures, many of which are interrelated. These include depopulation of rural areas, a reduction in participation in agricultural education, low levels of uptake in the farming sector (especially by young women), low incomes for farm businesses and poor health outcomes for farmers and others in rural areas [3]. Thus, despite the many benefits of industrialized agriculture, these farming practices can be seen to be contributing to declining rural employment and rural depopulation [22]. There is a need to consider alternative agricultural paradigms and transformative agricultural systems. Current conventional agricultural practice may not always be the best way forward for all regions in Australia.

These undesirable direct and indirect impacts of agriculture on the environment are becoming less acceptable to the global community and pressure is being put on governments to find alternative paradigms for food production.

### 1.4. Response from Agricultural Sector

The agricultural sector can respond quickly to change when required and has been seen to triple agricultural production over a 50-year period (1961–2011) due to the new technologies available during the Green Revolution [1]. These increases in production were mirrored by improved transport and post-harvest techniques which contributed to substantially longer value chains (farm gate to plate) as well as increased consumption of processed, packaged and prepared food.

Globally, there has been recognition that there is a need for a shift away from high-input, resource intensive farming to more innovative systems that can continue to increase productivity but at the same time protect and enhance the natural resource base [1,23–25]. A few of the many such alternative production systems include agro-ecology; agro-ecosytsems; agroforestry; climate smart agriculture; diversified farming systems; 'socially-modified' crops; sustainable intensification; and conservation agriculture [1,26–30]. The commonality between these approaches is that they are often multi-use, more holistic and, in many cases, built on indigenous traditional knowledge. Of the 250,000 plant species globally, 4% (20,000) have edible products (many from trees), but only 0.3% of edible plants are cultivated in agriculture [31], making plants a highly underutilized resource.

The Food and Agriculture Organization of the United Nations (FAO) [32] proposed four dimensions of food security, which include: reducing greenhouse gas emissions to limit and adapt to change; reducing impacts of different types of agricultural production on the world's ecosystems; developing rural areas to improve livelihoods and create jobs for poor people; and maintaining ecosystem services [4]. In this research paper, we describe an agricultural paradigm based on the commercial use of native foods which is in line with FAO proposed criteria [32]. We advocate for the use of a landscape ecological approach in understanding, evaluating and developing such a paradigm.

## 2. Alternative Agricultural Systems

Much of the discourse around climate change, vulnerability and food security, emphasizes cultivated foods, new animal breeds and crop varieties, and climate–crop modeling as the solution [2,5,33]. This is likely to be the main approach to meeting the challenges of global food security in the future. However, there are additional pathways that could also contribute with less impacts on the natural and cultural environment. This vision is reflected in recent times in affluent, western societies where rural change has transitioned away from a dominance of production values towards a variable mix of production and environmental protection values [34].

### 2.1. Agricultural Heritage Systems

Agriculture is defined by the Merriam-Webster dictionary as 'the science, art, or practice of cultivating the soil, producing crops, and raising livestock and in varying degrees the preparation and marketing of the resulting products'. There are, however, many alternative production systems that do not fit neatly within this definition. For example, the harvest of forest products often involves a degree of forest custodianship and management which contributes to the growth, quality and abundance of a harvested product. It could be argued that such practices should be considered as agricultural practice. Therefore, we will refer to some of these alternative practices as 'agriculture' in this research paper.

There are globally important agricultural heritage systems that have been developed by indigenous cultures over millennia [35]. These are often very complex, diverse and specific to local areas, involving techniques and practices that have contributed to community food security often in conjunction with conservation of natural resources and biodiversity. Agricultural heritage systems can still be found globally, with about 5 million hectares providing a vital combination of social, cultural, ecological and economic services to humankind [34]. An estimated 1.4 billion people manage such agricultural systems and landscapes globally, mostly family farmers, peasants and indigenous communities [34]. Many scientists acknowledge that traditional agricultural systems have the potential to provide solutions to the predicted changes and transformations facing humanity in an era of climate change, biodiversity loss and sociocultural issues [34].

### 2.2. Wild Foods

'Wild foods' constitute a niche area of food production that involve production and harvest with minimal impacts and interventions on the surrounding environment while at the same time providing incentives not to clear natural habitats. A 'wild food' can be described as an animal or plant product which is found in an undomesticated state in nature. Many of the commonly used products that the world relies on today have wild origins including most staple foods (corn, potatoes, tea, spices), medicines (aspirin, codeine), fibers (cotton, hemp), dyes (indigo and saffron), intoxicants (tobacco, opium) [35]. There is still a high demand from western markets for wild genetic plant stocks, with 25% of prescription drugs currently in use today having plant origins. Between 1981 and 2006, approximately 75% of new anticancer drugs were derived from plant compounds [36,37]. Ensuring future biodiscovery will require the conservation and management of the world's remaining natural habitats.

Non-Timber Forest Products' (NTFPs) are an example of a type of wild food. NTFPs were defined by FAO in 1995 as consisting of 'goods of biological origin other than wood, derived from forests, other wooded land and trees outside forests' [38]. Wild foods and NTFPs are an area of agriculture which contributes to millions of livelihoods worldwide. Globally, there are 300 million people living in predominantly forest ecosystems, with a large percentage of these people dependent on forests and their products for their livelihoods [39]. As such, NTFPs make up a considerable component of the world's food economy and are an important safety net during extreme events. These products are sometimes termed the 'hidden harvest' because their direct and indirect values are often not measured

nor included as part of official agricultural outputs [40]. NTFPs are collected for customary and commercial purposes, mostly managed sustainably by local people, communities and customary law.

Globally, many indigenous communities still have a high dependence on wild-collected plant products for their health, nutritional, cultural and spiritual wellbeing [39,41]. Agricultural and forager communities within 22 countries in Asia and Africa have been recorded as using an average of 90–100 species per location [5]. Much of the literature on food security emphasizes the production of cultivated foods, but clearly wild foods are making a substantial contribution to the global food basket [5]. Furthermore, many NTFPs are actively managed, which suggests there is a false dichotomy between agriculture and use of wild products.

Australian Aboriginal and Torres Strait Islander people (hereon Aboriginal people) are the custodians of the oldest culture on earth [42]. They continue to have extensive ecological knowledge and a deep, spiritual connection to their traditional lands [43]. Through customary care for and use of natural resources over tens of thousands of years, they have developed an intricate knowledge of the value of plant products [44,45]. A wide range of enterprises are emerging from this knowledge, including bushfood enterprises, native plant derived industries such as nurseries, seed harvesting, cut flowers, and a variety of botanical based medicinal and beauty products [46]. The resulting enterprises are largely based on wild harvest from traditionally managed estates, but also involve different models of cultivation such as enrichment planting and horticulture [47].

### 2.3. Niche Markets

Plant products play an important role in local, regional and international markets. At a local and regional level, they are often part of an indigenous customary harvest which trades, transports and sells products over vast distances along a diversity of value chains. In addition to market demand of wild plant material for specialized medical and pharmaceutical development, there is a rapidly growing consumer consciousness about links between health, diet and the environment, and an increased awareness of foods that are produced in safe, ethical and sustainable production systems [48]. This group of foods is referred to as "functional foods", which potentially have a positive effect on health above their nutritional values, in areas such as the prevention and management of health conditions [49]. Estimations of the revenue generated by the global functional food market vary considerably, however, it is estimated to have grown considerably over recent years. Market research estimates the global functional food market size as being 161.49 billion USD in 2018 and predicted to grow to 275.77 billion USD by 2025 [50].

Australia is well positioned to take advantage of this growing demand for functional foods. It has a very diverse endemic flora [51] with many species already having commercial applications in the fields of pharmacy, medicine, food, beverage, cosmetic, perfumery, and aromatherapy [52–56]. Coupled with this, Australian Aboriginal people have been using native foods for more than 40,000 years [57]. In recent years, there has been considerable interest among Aboriginal people in the commercialization of these products [46,58,59].

### 2.4. Sustainable Landscape Management

Aboriginal stakeholders are major landowners across northern Australia's tropical rangelands and have shown interest in a range of natural resource-based enterprise development opportunities [58,60–62]. The environments in which customary harvest practices take place are generally relatively intact ecosystems. Much of this area is under Aboriginal land tenure, is remote and has Aboriginal communities and Aboriginal Ranger groups actively involved in its management. If communities desire to scale up their customary use to commercial use, then they need knowledge of the impacts on the ecosystems in which they occur. This will require a greater understanding of the harvested species, the interconnectedness in the landscape and, more broadly, the ecology of the landscape relevant to the harvested species. Knowledge of important landscape patterns, processes and change will be fundamental in understanding and managing the dynamics of the systems in

which the species occur. Traditional Aboriginal ecological knowledge, coupled with sound harvest and scientific monitoring data, will be important information sources that can help to determine a culturally appropriate basis for establishing good management practices.

There are also large areas across the Australian Rangelands which are not in pristine conditions, having been impacted from frequent, intense wildfire, high densities of feral animals and modification from other land uses such as pastoral use, cropping and mining. This has resulted in soil erosion and biodiversity loss and, in instances, involved high levels of tree removal, altered water flows and introduced pastures [63,64]. These areas may require different consideration to those landscapes that are more ecologically and culturally intact. For example, priority considerations may include cultural and environmental restoration, alongside commercial priorities.

The discipline of landscape ecology has an important role to play in helping understand and inform the sustainable harvest for traditional and commercial use under changing climatic conditions. The focus of landscape ecology has largely involved spatial heterogeneity and ecological values with recent recognition that cultural values are also important elements in a landscape [65]. In turn, ecological and cultural knowledge informs the selection of appropriate business models. These models include wild harvest of natural resources, cultural values and practices, and remote rural economic settings. This integration necessitates a sustainable approach, addressing the triple bottom line.

Applying a landscape ecological framework can bring together different knowledge systems, values and priorities to measure the impacts and develop strategies for sustainable use without destroying the ecological integrity of the landscapes. Sustainability must include consideration of the socio-economic context of the communities harvesting the species. Integrated approaches help to understand the characteristics of species that have value, the markets that are likely to be interested in these characteristics, and the communities that harvest the species. Integrated, landscape approaches can also help to maintain important socio-ecological systems whilst providing for increased livelihood opportunities and allowing multifunctionality of land uses (conservation and development) [66]. However, landscapes are dynamic and can progress in different directions [67], especially under changing climatic conditions.

We posit that Australian Aboriginal people are well positioned to scale up their customary harvest of native foods to service rapidly growing, niche, functional food markets. For reasons discussed later, we suggest wild harvest as a suitable initial production model for supply of native plant products to niche markets, while ensuring benefits from resource use are retained in the landscape. There will be issues around sustainable use that need to be considered and management plans will need to be developed if they have not been already [68]. However, a landscape-focused framework will be most appropriate to measure landscape system health and impacts, as there are many overarching cultural, social, ecological and political factors that need consideration [65].

## 3. Case Analysis—*Terminalia ferdinandiana* (Kakadu plum) Enterprise

To demonstrate the potential of wild foods as an important contributor to food production, we take a case analysis approach. To explore the scaling up of customary harvest to commercial use, a case analysis of the wild harvest of the fruit Kakadu plum on Aboriginal owned traditional lands is reviewed. This case involves the wild harvest of a native, endemic fruit, *Terminalia ferdinandiana* Exell., by Aboriginal people across northern Australia. We discuss the species, its customary use, markets and production options available for scaling up from customary use, by small remotely located populations situated in a multifunctional savanna landscape.

A participatory research methodology was used in conducting this analysis, along with an ethnographic account of factors that have influenced the progress of this enterprise over the last 15 years [48,61,62]. The main method used for qualitative data collection was participant observations, which is a tool in many disciplines for collecting data about people, processes and culture [69]. A literature review using published and unpublished papers and reports was also used in gathering data to describe the north Australia Kakadu plum industry.

### 3.1. Properties

*T. ferdinandiana* is best known by the common name 'Kakadu plum' in Northern Territory (NT); *'gubinge'* in the Kimberley, Western Australia (WA), and many other Australian Aboriginal language names across its range. It will be referred to as 'Kakadu plum' in this paper as this is one of its most widely used common names. It is a member of the Combretaceae family [70], is endemic to northern Australia and is one of 200 species in the genus *Terminalia*, of which 29 species or subspecies are native to Australia [71]. *T. ferdinandiana* is a small to medium sized semi-deciduous tree that is found in the woodlands of the upper rainfall band of the Australian wet/dry tropics (see Figure 1). Its density is very variable over its range, but this species can occur in very high densities on or near the coast [61].

Kakadu plum is well known for its phytochemical properties. It has the highest vitamin C (ascorbic acid) of any fruit in world [72]. These exceptionally high levels of vitamin C were first detected in 1982 through a study of the nutritional composition of bushfood used by Australian Aboriginal people [73,74]. The fruit and leaves also have very high levels of ellagic and gallic acid and other polyphenolic compounds. These, along with the vitamin C, provide high antioxidant values which are known to reduce risk of diseases such as cardiovascular disease, cancer, stroke, and rheumatoid arthritis [72,75–79]. These phytochemicals have also been proven to have high antimicrobial properties [80]. These phytochemical properties, and their demonstrated applications, have created a commercial demand from the food and beverage, pharmaceutical, cosmetic and nutraceutical industries [81].

### 3.2. Customary Use

Traditional foods continue to be an important part of the diets of Aboriginal people. A study of five Aboriginal communities in the NT, Australia, found that 89% of the people interviewed consumed a variety of traditional foods fortnightly [82]. Aboriginal people have a strong affiliation with the Kakadu plum and many Aboriginal language groups across its range have a close cultural connection and varying uses for this species and its products [45,83]. The fruit has been recorded as being consumed for quick energy and as a refreshment on hunting trips and is used for a variety of other medicinal purposes, such as treating colds and congestion [44,45,84,85]. The inner bark is used to treat skin disorders and as well as fungal infections such as ringworm [61].

### 3.3. Kakadu Plum Markets

The ongoing research and biodiscovery of different phytochemicals in Kakadu plum and the identification of potential commercial applications has stimulated several market sectors. However, until recently, market signals have been very inconsistent between years, which has contributed to the formation of a disjunct and poorly coordinated supply network. Response to these market signals for the supply of Kakadu plum has come from both Aboriginal and non-Aboriginal people, and involved three main production systems, namely horticulture, enrichment planting and wild harvest [47,48,86]. We describe these production systems below.

### 3.4. Kakadu Plum Production Systems

Wild harvest is the production system that most Aboriginal people (cooperatives, communities, family groups, and individuals) across northern Australia are involved in, through several different business structures. Kakadu plum has been commercially harvested from the wild in the Northern Territory (NT) since 2005 and was initially trialed through several Aboriginal Ranger groups [61]. One of these groups, the Thamarrurr Rangers from the Thamarrurr Region, NT, hosted a Kakadu plum enterprise and acted as the consolidator, by managing the fruit collected and linking with markets (Figure 1). The Thamarrurr Ranger's primary responsibility is natural resource management rather than commercial development, so after a few years, they handed over the consolidator role to a local Aboriginal owned and operated business, the Palngun Wurnangat Aboriginal Corporation

(PWAC). In 2020, approximately 15 years since the Kakadu plum collection trial first started, the Thamarrurr Kakadu plum Enterprise still operates as a community owned and operated business. It now engages PWAC and the local Aboriginal development corporation, Thamarrurr Development Corporation, to assist in supporting operational activities. Annually, this community enterprise has purchased tons of wild harvested fruit from the community members who wild harvested it from their traditional estates [87]. This enterprise provides significant monetary and non-monetary benefits for the community.

Another Ranger Group in the NT, the Bawinanga Rangers, who are supported by the Bawinanga Aboriginal Corporation (BAC) in Maningrida, Central Arnhem Land (Figure 1), were also involved in a small, wild harvest of Kakadu plum between 2005–2008. BAC is currently trialing harvest of a range of native bushfoods for sale to restaurants and other markets around Australia [88] and hope to expand this activity to include some of the 32 Aboriginal clan estates in their jurisdiction.

The Kimberley area of WA is another geographical area where Aboriginal people wild harvest Kakadu plum commercially. Some examples of Aboriginal owned and operated businesses operating in this area include: Twin Lakes Cultural Park, Kimberley Wild Gubinge, Lombadina Community and Mayi Harvests. Twin Lakes Cultural Park is a family business located north of Broome on the Dampier Peninsula (Figure 1). The Aboriginal traditional owner, Bruno Dann, is a Nyulnyulan person who lives on his traditional land making a living from wild harvest of Kakadu plum and cultural tourism. He sells his fruit to a non-Aboriginal company, Living Earth Pty Ltd., which uses the Kakadu plum powder as an ingredient in chocolate [89]. Kimberley Wild Gubinge is another Aboriginal owned business which is situated north of Broome in the Dampier Peninsular, WA (Figure 1). It purchases fruit from local harvesters which it processes into powder in its solar-powered premises, providing local livelihoods and preservation of local knowledge though resource use and appropriate land management [90]. The Lombadina Community and Lombadina Aboriginal Corporation, established in 1985 on the Dampier Peninsular, in the Kimberley Region of WA, have been in partnership with traditional owners and communities to pick and sell wild harvested Kakadu plum and have recently started growing Kakadu plum in orchards [91]. Mayi Harvests was established in 2006 and is situated in West Kimberley on traditional family lands in Ngumbarl and Jabirr-Jabirr country. They harvest several native plant species, including Kakadu plum (locally called *gabiny*) which they sell as both dried and frozen products [92].

Enrichment planting is another production system in which Kakadu plum trees are grown in the Kimberley area of WA. Enrichment planting involves additional planting into wild populations using wild harvested seed stock of desirable properties to increase tree density [47]. The Kimberley Training Institute (KTI) in Broome, WA, has established an enrichment planting trial at its Balu Buru site, in partnership with WA Department of Conservation and Land Management [47]. KTI is a horticultural training provider which uses the Balu Buru trial site in their training. KTI also supports Aboriginal communities to establish other Kakadu plum enrichment plantings and orchards. The Balu Baru site has over 1000 Kakadu plum trees that have been enrichment planted over a 5 year period. They are developing a 'Savannah Enrichment' approach which combines traditional burning practices with modern horticultural techniques. These trials being conducted at Balu Buru provide valuable lessons in the propagation and growth of Kakadu plum in competition amongst the dense acacia thickets which have developed because of unmanaged fire regimes. They indicate that at a landscape scale, savanna enrichment practices could help change the structure of plant communities and reduce the fuel loads and occurrence of higher intensity wildfire. In turn, this would reduce the coinciding biodiversity loss while at the same time providing livelihood opportunities for local people and an economic incentive to manage the landscape differently [47].

Kakadu plum is also grown as a monoculture in horticultural settings. Kakadu Life Pty Ltd. is the main non-Aboriginal company that grows Kakadu plum at scale. Their production is based in NT, Australia, with distribution based in Perth, WA, and they sell a variety of Kakadu plum products, many with organic status [93]. There are also several Aboriginal owned communities that grow Kakadu

plum in the Kimberley area of WA in a horticultural setting. These include WA's largest Aboriginal community, Bidyadanga Aboriginal Community, situated 180 km south of Broome, which started growing Kakadu plum almost 20 years ago [94] (Figure 1). Mamabulanjin Aboriginal Corporation is an Aboriginal Resource Centre based in Broome and is another group that is growing Kakadu plum in a horticultural setting [95]. More recently, GoGo Station Pty Ltd., an Aboriginal pastoral station situated near Fitzroy Crossing, WA, set up a trial plot of 200 Kakadu plum trees under drip irrigation [96].

The Kakadu plum industry across northern Australia is established and growing. In summary, the exceptional phytochemical properties of Kakadu plum, the commercial applications and market demand, and knowledge from generations of customary use, underpin an established and growing Kakadu plum supply chain. The main model of production of Kakadu plum is wild harvest from Aboriginal traditional estates, supported by local Aboriginal corporations. Value chains involve both Aboriginal and non-Aboriginal actors within or outside of these estates. In many cases, enterprises have been developed by local Aboriginal people who have customarily harvested this species but who have now expanded these practices and incorporated business practices for commercial use. Enrichment and horticultural plantings are emerging as new modes of supply for this species, with the support of regional training institutions. Inspection of the known distribution of *T. ferdinandiana* would suggest potential for wider uptake of the commercial use of Kakadu plum, particularly in the NT (Figure 1).

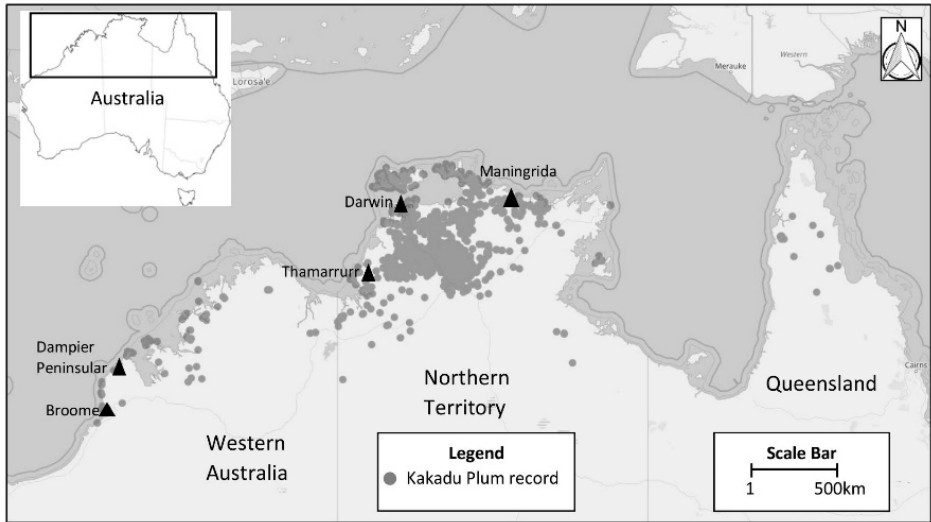

**Figure 1.** The distribution of the endemic *T. ferdinandiana* in Western Australia (WA), the Northern Territory (NT) and Queensland (QLD). Triangles represent place names and dots represent herbarium records of *T. ferdinandiana,* from Atlas of Living Australia [97].

## 4. Discussion

We now discuss the important considerations in wild harvest for customary purposes as well as commercial use and the scaling up to meet larger commercial demands for supply. We will also discuss the use of a landscape ecology approach for valuing, understanding and developing use of native foods for commercial purposes.

### 4.1. Customary Sector and Wild Harvest

Around Australia, Aboriginal people have used a variety of methods of food production prior to colonization. These range from fire being used to modify vegetation structure and composition for wild harvest purposes [98] through to more conventional types of agricultural practices including domesticating plants, sowing, harvesting, irrigating and storing crops, and implementing aquaculture and other farming practices [99–101].

The current Aboriginal economy has also been described as a 'hybrid' economy comprising of three sectors: customary, state and market [102]. The customary sector refers to subsistence harvest for food and cultural purposes; the state includes social security and government-funded programs such as 'work for the dole' schemes; and the market sector relates to the free market, most notably the fine arts and craft industry established in Aboriginal Australia. The customary sector constitutes a range of productive activities that are based on cultural continuities and cultural identity, such as hunting, fishing, gathering of bushfood, art and craft production, caring for kin and caring for country [103]. The Aboriginal economy has changed over time [104], but customary harvest is still a very important component of Aboriginal livelihoods. It is based on an intricate ecological knowledge and connection with their traditional lands. Aboriginal people relate well to the idea of a 'culture-based economy' which incorporates their knowledge, connection to their traditional estates and epistemology [105]. A culture-based economy involves a multifunctional approach and operates at a landscape scale.

Wild harvest of bushfoods for commercial purposes can be viewed as an extension of customary practice. It relies on Aboriginal ecological knowledge and practices, often still within cultural perimeters, but aims to service larger and external markets. As the scale of harvest increases, there will be concerns around sustainability and impacts of harvest that will need to be addressed, alongside the many other economic, ecological, social, and cultural benefits that come from this activity. Even though Kakadu plum is an abundant species and has an extensive natural distribution, the impacts of increasing harvest levels for commercial use, in conjunction with changing climatic conditions, will require careful land management practices. In response, the Northern Territory Government has developed a management program for Kakadu plum (*Terminalia ferdinandiana*) to ensure its sustainable use [68]. However, there are several case studies that demonstrate that Aboriginal people have established production systems which utilize wildlife for both customary (non-market) and commercial purposes, including use of saltwater crocodiles, long-necked turtles and raw materials for artworks [59,106–108].

Tropical savannas across northern Australia need active land management for their natural and cultural values to remain intact. This requires people to be living on country in the savannas, with knowledge of that country and how to manage it (such as traditional burning practices). People also need livelihoods to be able to thrive in these remote places across the savannas. However, except for the Aboriginal Ranger Program, there are very limited employment and enterprise opportunities in remote Aboriginal townships. The economic status of Aboriginal people is the lowest of any demographic group of Australians [109,110], with unemployment rates being as high as 90%, if various government welfare programs were not taken into consideration [111]. Commercial use of natural resources offers livelihood opportunities for Aboriginal people who have expertise in both customary use and land management as well as the right and a strong desire to be involved [61,62,112].

*4.2. Scaling up from Customary Harvest*

Extending harvesting from customary to commercial purposes is a manageable transition for Aboriginal harvesters because they have ecological knowledge about the resource (when it should be picked, where the best picking sites are, landscape management requirements, etc.,) and know the cultural protocols in which the resource must be harvested (access and harvest permissions, cultural sensitivities or prohibitions etc.). However, the component of commercialization that some Aboriginal communities might find difficult is building an appropriate business structure.

There are many complexities and challenges in developing an Aboriginal business, especially in remote and regional localities within the savanna landscapes [48,113–115]. Business development in remote Aboriginal communities is different to that in other Australian communities [116]. Enterprise development in Aboriginal communities is often funded through government programs and is likely to have originated without commercial intent and often involves subsidized community-based activities [117]. Many of these enterprises are largely focused on social goals in the absence of economic criteria for success. These enterprises often lack the business acumen required to make decisions that lead to viable long-term businesses in this distinctive landscape context [48]. This confusion

between social and economic objectives has been cited as an important contributor to business failure in Aboriginal businesses [87,118]. There are, however, many examples of Aboriginal community-based enterprises (such as those described in Section 3.3 above) which have managed social, cultural and economic priorities successfully.

For Aboriginal people to have greater control of the Kakadu plum industry, there is a need for them to take a 'whole of industry' approach and become leaders in all aspects of the business, including research, harvest, processing and marketing. The Indigenous Land and Sea Corporation is working with Aboriginal people to achieve this goal. In 2018, it established the Northern Australian Kakadu Plum Alliance (NAAKPA). NAAKPA currently consists of a consortium of eight Aboriginal owned enterprises which ethically harvest and process Kakadu plum across northern Australia [119]. This Alliance provides support to its members as they grow their businesses while at the same time providing stability and reliability to the Kakadu plum supply chain. Such cooperative or collective ventures are likely to play an important role in the development of savanna enterprises, in so far as they support a focus on Aboriginal economic development as well as ecological, social and cultural priorities. A cooperative model across several savanna sites will also help manage the risks to supply, inherent in wild harvest.

As markets develop and demand for Kakadu plum increases in the future for Kakadu plum there may be a need for greater uptake of alternative production systems to complement wild harvest. This will place greater emphasis on domestication of Kakadu plum to meet demand. Enrichment planting is one alternative form of domestication which has been described earlier in this paper (Section 3.4) [47], which could contribute to more consistent yields and greater volumes of supply. More conventional horticultural production systems may also need be considered. Domestication may be desirable for some Aboriginal producers to meet the demands of larger markets. It is often seen as comprising of a spectrum of increasing levels of human intervention in the production of a species for human benefit [120]. There are some very relevant resources documenting the process of domesticating culturally important, indigenous food-tree species over a 25-year period in tropical/subtropical Africa [25]. This body of applied research is focused on domesticating trees and creating multifunctional landscapes which can reverse the cycle of land degradation and its associated social deprivation issues [26].

Leakey [121] describes food species as falling into four categories: i. Internationally important and widely cultivated staple foods, ii. Widely cultivated case crops, iii. Locally domesticated and cultivated species or 'orphan crops', which also have wider potential, iv. Culturally important species used for customary use and little known outside their natural range. He suggests that the first three categories have made the transition from ethnobotany to agriculture hundreds, if not thousands, of years ago, while the fourth category is currently making that step following recent research. Despite being focused more on agroforestry than wild harvest, this long-term participatory research project has demonstrated some interesting findings that could be incorporated into Australian Aboriginal agricultural development. This is particularly the case for a tree species like Kakadu plum with so many valuable attributes—edible fresh or processed fruit, a source of vitamin C and rich in antioxidants. The identification of 'ideotypes' to capture ideal phenotypic trait combinations for different end products and associated markets [122], can differentiate suppliers across the savanna landscape. Taking a geographically decentralized approach to reduce risk in narrowing the genetic base of species [123] is an approach that would be very applicable to Kakadu plum, as there are many varieties across its range on numerous traditional estates and there may be cultural reasons for land owners wanting to keep genetic strains represented by 'ideotypes' separate. Creating Rural Resource Centers to assist in technical training and business support [25] and making partnerships which link domestication and commercialization programs, is considered critical to the success of commercialization and demonstrates the long-term level of support and commitment required to progress the development of community-based enterprise development. Examples of successful Kakadu plum enterprises developed to date demonstrate this.

Given there are also many degraded landscapes across the Australian rangelands, there may be potential to establish an agroforestry domestication program. Processes to domesticate native species need to protect the interest of the traditional estate owners in Australia. Approaches, such as the *sculptured seedling technique* to revegetation, which relies on a knowledge and understanding of the natural vegetation in areas to match site capability with appropriate species, may prove useful [124]. Other processes, such as utilizing socially modified crops rather than genetically modified crops, are more likely to prevent the loss of genetic variability from the landscape and protect local interests and benefits [25,26]. This process could, in time, supplement yields from wild harvest and help provide the volume and consistency of supply to secure relationships with larger markets. This work will need to be done in close consultation with traditional landowners as there will be many issues (access, identifying desirable phenotypic traits, cultivar development, cultivation techniques) that will require their participation and customary authority.

### 4.3. An Intregrated Landscape Approach to Management of Country

Aboriginal people own vast areas of land across northern Australia. In the NT alone, Aboriginal people make up around a third of the population and own over half of the land, mostly under a communal title. Much of the natural range of Kakadu plum is found on Aboriginal lands across the northern Australian tropical savannas, which still consist of relatively intact landscapes. These lands are managed by a mixture of traditional Aboriginal and western land management practices, in collaboration with the Aboriginal Ranger groups and in conjunction with traditional land management practices or authority [125]. Landscapes managed by traditional land management practices in tropical northern Australia have been shown to have greater ecological integrity than those managed in the absence of Aboriginal land managers and their traditional ecological knowledge non-Aboriginal managed sites [126].

Access to native plant resources on Aboriginal land, their commercial use and sustainability, are regulated through customary lore and state legislation. Traditional owners, who are the designated decision makers for individual clan estates, have cultural obligations to look after their country and this relates to caring for both natural and cultural resources and maintaining their spiritual connections. People are integrally linked to place and place is integrally linked to people [127]. In the NT, the *Aboriginal Land Rights Act* (NT) 1976 states that any commercial activity must be approved by the traditional owner(s). Permits to access and commercially harvest on Aboriginal lands must be with the authority of the traditional owner(s) and must be captured in a Land Use Agreement between the Aboriginal Land Trust, on behalf of the traditional owner(s), and the proponent. Sustainable use of native plants and animals is regulated through the NT Department of Environment and Natural Resources through a permit system and has long been a focus and part of the NT Government's conservation strategy [128]. The sustainability of commercial use of Kakadu plum is of utmost importance to the NT Government and a 'Management Plan for *Terminalia ferdinandiana* in the Northern Territory 2018–2022' is in place to ensure wild populations and the species habitat are adequately maintained across the NT of Australia [68].

There is a risk that over time the benefits of commercial use of native plants could be realized off Aboriginal owned lands and to the exclusion of indigenous peoples. Kakadu plum is an industrial crop and there has already been an incident in 2004 where two multinational companies have tried to export Kakadu plum tissue culture out of Australia without permission or benefit-sharing agreements [85,129]. However, there are two key ways that the interests of Aboriginal peoples can be protected. Firstly, by targeting premium markets that value culturally identified and ethically sourced products. Secondly, there are legislative mechanisms in place to protect interests and ensure benefit sharing with landowners, including traditional landowners and traditional knowledge holders [130]. Australia is a signatory of the 'Convention on Biological Diversity 1992', under which the 'Nagoya Protocol' on 'Access to Genetic Resources' and the 'Fair and Equitable Sharing of Benefits Arising from their Utilization' has been framed to protect the interests of indigenous peoples and communities.

In the Northern Territory, Australia, land held under Aboriginal Freehold title, and awarded under the *Aboriginal Land Rights (Northern Territory) Act* 1976, requires special land use agreements with traditional landowners before parties can access or use natural resources from this land. Finally, Australian states and territories, including the three jurisdictions in which Kakadu plum occurs, have biodiscovery acts and regulation to manage the accessing, collection and transfer of biological materials collected, and the benefit that flows from their use.

Australian Aboriginal people are the custodians of the oldest continuous culture on earth [42] and have a deep, spiritual connection to their 'country' [43]. Tens of thousands of years of Aboriginal land management can be described as 'sustainable' in that it has resulted in a productive and sustaining relationship between humans and their environment [100]. An integrated landscape approach seeks to understand the relationship between diverse values, which requires a transdisciplinary approach, that incorporates the ecological, economic, social, and cultural considerations with people from diverse cultural, educational and philosophical backgrounds [65].

## 5. Conclusions

The agricultural sector faces significant challenges now and into the future, with changing climates and a rapidly growing global population. Historically, indigenous agricultural systems, in their many forms, have accounted for the food security and livelihoods of many millions of people globally. In Australia, wild harvested foods continue to make an important contribution to Aboriginal livelihoods, health and wellbeing, and provide economic opportunity where remoteness, education and infrastructure allow few alternatives [131].

Aboriginal Australians are major landowners across northern Australia with strong cultural connections and intricate knowledge of their land and the plants and animals within. Many groups still rely on customary harvest for their livelihoods and use these products for a diverse range of nutritional, medicinal and cultural purposes. A north Australian native plant, *T. ferdinandiana*, which provides customary food and medicine, is currently being commercially wild harvested, enrichment cultivated and horticulturally grown by Aboriginal people. We conclude that a scaling up of customarily harvested products, such as Kakadu plum, is both desired by Aboriginal people and an appropriate alternative agricultural paradigm to meeting high value niche market demands, thus contributing to global food security by broadening the base of food species used and valued. As markets grow, domestication may become necessary. Models of domestication that promote community participation and local benefit and that protect the natural genetic diversity of a species across its landscape should be prioritized. Investment in regional training is required to support this process [25,121,132]. In parallel with the scaling-up of customary harvest, domestication may allow for greater consistency and volume of supply leading to greater market confidence and more competitive pricing structures.

Wild harvest can be conducted with minimal impact on the surrounding environment and the many opportunities that exist for scaling up the supply of native plant products that have market demand, must be explored. This will require further surveys and research to identify commercial potential through bioprospecting and incorporation of Aboriginal knowledge. There are already many species being harvested for customary purposes by Australian Aboriginal people that may have commercial potential [4,61,62]. We posit that globally there are many other similar customary harvested products found on other natural landscapes that could be harvested sustainably to meet market demands and contribute to food security.

This approach to food production would be best managed at a landscape scale. This would draw on concepts from landscape ecology to ensure sustainable production and protection of the multifunctionality of landscapes, through the conservation of all the important landscape values (ecological, cultural and social). North Australian landscapes are expansive and largely intact due to their poor soils, remoteness from markets and highly seasonal rainfall. There have been several failed attempts to 'develop the north' with the planning being largely influenced by external factors with little regard to Aboriginal aspirations [65] or local conditions [63]. Landscapes across northern Australia

have been managed for thousands of years by Aboriginal custodians and are rich in both natural and cultural heritage values. Such holistic and integrated landscapes should not be compromised by inappropriate agricultural models. Aboriginal custodians have obligation to 'care for country' and in most cases, these obligations exclude large scale clearing and development. An alternative development practice is a 'cultural approach' to land use which is holistic in nature, building on customary lore and practice to service new and expanding markets that value sustainable practice and organic, ethically sourced foods.

This idea of holistic planning for multifunctional landscapes is not new and has been supported by several authors [65,131,133]. Valuing ecosystem services, Aboriginal knowledge, economic values of native plants, and managing for multiple values, are common threads of an integrated landscape approach. Many of the landscapes across northern Australia are "undeveloped", have relatively intact ecosystem services, along with people who have maintained cultural connectedness, albeit to varying degrees. This is an ideal time to prevent loss of important multifunctionality by taking an integrated and holistic landscape approach to land management and economic development in this region [134]. This can help to capture the ecological, economic, social and cultural values that are inherent in these landscapes and plan a shared vision for the future that provides sustainable livelihoods for current and future generations resident in these northern landscapes [135]. This vision of landscape as an agricultural system also prioritizes the retention of benefit locally, in an economically poor but culturally and biologically diverse landscape. We need to effectively conserve ecological and cultural integrity whilst creating novel future farming systems while we have them, rather than trying to apply these principles to landscapes which have been altered and no longer have Aboriginal cultural connections.

This paper has introduced the concept of an agriculture paradigm which is compatible with the concept of a 'customary economy' that Aboriginal people aspire to develop [94,136]. Future agribusiness models in these landscapes should build on the existing customary knowledge to create an innovative and appropriate agricultural paradigm which will contribute towards Aboriginal livelihoods and global food security issues in a changing world.

**Author Contributions:** Conceptualization, J.G., D.P. and P.W.; methodology, J.G.; investigation, J.G.; writing—original draft preparation, J.G.; writing—review and editing, D.P. and P.W.; supervision, P.W. All authors have read and agreed to the published version of the manuscript.

**Funding:** This research received no external funding and in part of a PhD.

**Acknowledgments:** Charles Darwin University for hosting this PhD research. The people and organizations in the Thamarrurr Region that have shared their experiences over many years. Chris Brady for his supervision and providing comments on a draft.

**Conflicts of Interest:** The authors declare no conflict of interest.

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
