# Peer review of "Old Ways, New Ways—Scaling Up from Customary Use of Plant Products to Commercial Harvest Taking a Multifunctional, Landscape Approach"

_land, doi:10.3390/land9050171_

Round 1
Reviewer 1 Report
In general the study was interesting and informative. Better definition of the terms and a little more information regarding the culture of the potential crop species. While I agree with the necessity and scope of the approach to a more traditional 'agricultural' system, and its importance to Indigenous peoples, there is a lack in the completeness of the discussion of the potential of such a system. The authors begin by describing a traditional system and its intrinsic value to a reduction of the "wicked" problems of agriculture, and then near the end of the paper talk about the potential need to cultivate this crop in conventional orchards, once a market has been developed.Would this not take a potentially ecologically sustainable crop and take it into the "wicked" realm?
I would like to see greater emphasis on policy options to help protect this type of industry both at the territorial, national and international (FOA??) level. It would appear that this could go from food security crop, to an industrial crop (cosmetics, perfumes, functional food additives) where the value developed by the pioneers of the crop will be realized by others and with the potential of production to leave these traditional lands.
Suggested that domestication may be needed. Why? How best carried out to protect genetic diversity?
what is enrichment plantings? how does this impact the diversity of the species? seed propagated? cuttings? is the species self or cross pollinated? how would this be affected by the system of culture (seed vs. vegetative propagation?)
Areas of the discussion that were omitted: What about the isolation of the production sites from 'new' markets?; Infrastructure issues and ecological impact of this; and finally, the adaptation of this type of system to climate change (first sentence of the Introduction).
There are a number of other comments and minor corrections to be found in the attached pdf.

Author Response
Dear Reviewer 1,
Many thanks for taking the time to review my paper and provide such valuable comments. Please find my responses to your queries and comments below and in that attached pdf.
Regards, Julian
- In general the study was interesting and informative. Better definition of the terms and a little more information regarding the culture of the potential crop species. While I agree with the necessity and scope of the approach to a more traditional 'agricultural' system, and its importance to Indigenous peoples, there is a lack in the completeness of the discussion of the potential of such a system. The authors begin by describing a traditional system and its intrinsic value to a reduction of the "wicked" problems of agriculture, and then near the end of the paper talk about the potential need to cultivate this crop in conventional orchards, once a market has been developed. Would this not take a potentially ecologically sustainable crop and take it into the "wicked" realm?
Author response: Not necessarily. There is a spectrum of potential production systems, in addition to wild harvest. This spectrum includes enrichment plantings (Lee et al 2016) or types of agroforestry and mixed planting. These other models may prioritise other values, such as environmental or cultural values, as well as production volume, but not only production volume.
- I would like to see greater emphasis on policy options to help protect this type of industry both at the territorial, national and international (FOA??) level. It would appear that this could go from food security crop, to an industrial crop (cosmetics, perfumes, functional food additives) where the value developed by the pioneers of the crop will be realized by others and with the potential of production to leave these traditional lands.
Author response: (lines 578-594) Yes there is a risk of the benefits of commercial use of native plants to be realised off traditional lands and to the exclusion of indigenous peoples. Kakadu Plum is already an industrial crop and there has already been an incident where an international company tried to take germplasm out of Australia to establish production elsewhere. However, there are two key ways the interests of indigenous peoples can be protected. Firstly, is targeting premium markets that value culturally identified and ethically sourced products. Secondly, there are legislative mechanisms in place to protect interests and ensure benefit sharing with landowners, including traditional landowners and traditional knowledge holders (Robinson et al 2017. Australia is a signatory of the Convention on Biological Diversity 1992, under which the Nagoya Protocol on Access to Genetic Resources and the Fair and Equitable Sharing of Benefits Arising from their Utilization to the Convention on Biological Diversity been framed to protect the interests of indigenous peoples and communities. In the Northern Territory, Australia, land held under Aboriginal Freehold title, and awarded under the Aboriginal Land Rights (Northern Territory) Act 1976, requires special land use agreements with traditional landowners before parties can access or use natural resources from this land. Finally, Australian states and territories, including the three jurisdictions in which Kakadu plum occurs have biodiscovery acts and regulation to manage the accessing, collection and transfer of biological materials collected, and the benefit that flows from their use.
Gorman, J., Wurm, P.A.S., Vemuri, S., Brady, C., Sultanbawa, Y. (2019) Kakadu Plum (Terminalia ferdinandiana) as a Sustainable Indigenous Agribusiness. Economic Botany in print. doi.org/10.1007/s12231-019-09479-8
Cunningham, A.B.; Courtenay, K.; Gorman, J.T. and Garnett, S. 2009. Eco-enterprises and Kakadu Plum (Terminalia ferdinandiana): “best laid plans” and Australian policy lessons. In: Economic Botany 63(1): 16-28.
Robinson, D. F., Abdel-Latif, A., & Roffe, P. (Eds.). (2017). Protecting Traditional Knowledge: The WIPO Intergovernmental Committee on Intellectual Property and Genetic Resources, Traditional Knowledge and Folklore. Taylor & Francis
- Suggested that domestication may be needed. Why? How best carried out to protect genetic diversity?
Author response: (lines 510-513): Domestication may be desirable for some Aboriginal producers and to meet the demands of some markets, where domestication is seen as comprising a spectrum of increasing levels of human intervention in the production of a species for human benefit (Zeder 2006) and in participatory domestication described by Leaky and colleagues (Leakey 2019).
Gorman, J., Wurm, P.A.S., Vemuri, S., Brady, C., Sultanbawa, Y. (2019) Kakadu Plum (Terminalia ferdinandiana) as a Sustainable Indigenous Agribusiness. Economic Botany in print. doi.org/10.1007/s12231-019-09479-8
Leakey, R. R. (2019). From ethnobotany to mainstream agriculture: socially modified Cinderella species capturing ‘trade-ons’ for ‘land maxing’. Planta, 1-22.
Zeder, M. A. (2006). Central questions in the domestication of plants and animals. Evolutionary Anthropology: Issues, News, and Reviews: Issues, News, and Reviews, 15(3), 105-117.
- what is enrichment plantings? how does this impact the diversity of the species? seed propagated? cuttings? is the species self or cross pollinated? how would this be affected by the system of culture (seed vs. vegetative propagation?)
Author response: Enrichment planting is discussed in ‘3.4 Kakadu plum production systems’ of this paper – line 381-396. There is a paper which is all about the enrichment planting of Kakadu Plum in the Kimberley.
Lee, L. S.; Courtenay, K. Enrichment plantings as a means of enhanced bush food and bush medicine plant production in remote arid regions: Are view and status report. Learning Communities: International Journal of Learning in Social Contexts, Special Issue: Synthesis and Integration 2016, 19, 64–75.
- Areas of the discussion that were omitted: What about the isolation of the production sites from 'new' markets?; Infrastructure issues and ecological impact of this; and finally, the adaptation of this type of system to climate change (first sentence of the Introduction).
Author response: The impact of commercial harvest of Kakadu Plum is something that needs consideration. It is a very common species that has a natural distribution of a vast area so if managed properly the sustainability and ecological impact can be regulated. The NT Government has a management plan specially to do this:
Gorman, J., Brady, C., Clancy, T. (2018). Management Program for Terminalia ferdinandiana in the Northern Territory of Australia 2018-2022. Northern Territory Department of Environment and Natural Resources.
Added text (lines 454-458) in the text in response to this comment.
Author response: no action taken - comment only
Tropical savannahs will get impacted by climate change but the great thing about wild harvest is that you don’t invest huge amount of money as you do in cultivation. Perhaps the distribution of Kakadu Plum will change with climate change, but so will the practices of the harvesters – along with harvest practices of other species for customary and commercial purposes.
- There are a number of other comments and minor corrections to be found in the attached pdf.
Author response: these have been addressed in the pdf.
Submission Date
21 April 2020
Date of this review
25 Apr 2020 23:01:

Reviewer 2 Report
I think this makes a nice case for a follow-up study on how able this landscape is to contribute to food production. Obviously, this will not end dependence on wheat, rice, and maize production no matter how intensely the harvest is managed.
I would like to see the intro scaled down. It does not need to be near as long and depressing. There are minor editorial comments in the attached PDF.

Author Response
Dear Reviewer 2,
Many thanks for taking the time to review my paper and provide valuable comments. Please find my responses below and in the attached pdf.
Kind regards,
Julian
I think this makes a nice case for a follow-up study on how able this landscape is to contribute to food production. Obviously, this will not end dependence on wheat, rice, and maize production no matter how intensely the harvest is managed.
Author response: I agree, this type of agricultural practice will only be a relatively small contributor. I do mention this at the start of section 2.0 Alternative agricultural systems (lines 149-155 in the revised version of the paper).
I would like to see the intro scaled down. It does not need to be near as long and depressing. There are minor editorial comments in the attached PDF.
Author response: I have restructured the introduction as per suggestion of Reviewer 3 – it is now less long and depressing.

Reviewer 3 Report
Review of land-794877: Old ways, new ways - scaling up from customary use of plant products to commercial harvest taking a multifunctional, landscape approach.
General comments:
I found this to be a very well written research paper on a very interesting topic. It is unfortunately spoiled by Section 1, which doesn’t meet the high standards of the rest of the paper. I really had the feeling that the introduction was hastily tacked together so that the authors could get onto the business of writing the real paper.
In section 1, I don't see the structure of the text that is presented. You bring up a lot of concepts, and they are all valid, but they are mixed and scattered throughout the text. For example, social impacts are described in lines 116 to 120, and then again in lines 141-148. Biodiversity loss comes in lines 42, 97, 102, 113, and 121. If you are going to include so many concepts, which is perfectly legitimate, the structure is really important and I think you need to restructure Section 1 entirely.
I don't think this is as big a job as it sounds. All the material is there. So, I'd describe it as a 'minor revision'
A possible way to do it would be to first talk about the need for production: food security for a growing population in a changing climate. Then talk about how agriculture is addressing this challenge by increasing production. Then talk about the associated problems with the increased production: carbon output, biodiversity loss, etc. one at a time. And then discuss how agriculture is searching for, and applying, solutions to these problems. There are of course other structures, you could use but it has to be clear.
From section 2 onwards, the paper was a pleasure to read.
There are a couple of typos here and there, so a quick read through to find and correct these would be worth the effort.
Specific comments:
Line 15: Accordance means 'agreement with'. Probably not the best word in this case.
Line 17: ‘Governments’ doesn’t need a capital letter
Line 48: should be a semi-colon.
Line 49: should be 'maintaining'.
Line 53: here is a natural break in the text, so probably needs a subheading.
Line 67: strange wording. It suggests health is an agricultural product.
Line 68: This sentence comes a bit out of nowhere and doesn't say much. I suggest putting it with the other sentences that discuss the effects of global warming.
Line 70: The sentence starting with ‘The impacts’ should actually be part of the first sentence of the previous paragraph.
The paragraph starting at line 72: I think you should move this paragraph to the beginning of this new section. First, state why we need to produce more food and then explain how climate change is making that difficult. To me, that would be more logical.
Lines 92-94: awkward sentence.
Line 94-98: awkward sentence. The word 'and' comes 6 times.
Line 100: Obesity rates: That may be, but what has it got to do with agricultural outputs?
The paragraph from lines 99 to 120: I don't understand why this paragraph is included under the sub-heading 'response from agricultural sector'. To me, it sounds more like part of the 'problem statement' section
Line 121: There have been some statements already that agriculture is responsible for biodiversity loss. This sentence should probably go somewhere near those.
Lines 123-125: This sentence is a bit lost here.
Line 126: Is perverse the right word?
Line 132: typo in the word ‘systems’
Line 179: I'd stick with biodiversity loss and socio-cultural issues. There are plenty of problems to solve there without introducing energy and financial crises.
Line 198: 1.6 million sounds a bit low to me. I think it might refer to the number of people dependent on forests and their non-timber products for their livelihoods
Line 199: I'm not sure you can say ‘huge’. I believe there are around a billion people in the world who depend on agriculture for their livelihoods.
Line 205: I think you need to specify ‘wild-collected plant products’ rather than just plant products.
Lines 265-266: Awkward sentence.
Lines 621-624: It's probably better to put all the recommendations in the conclusions section.
Author Response
Dear Reviewer 3,
Many thanks for taking the time to review my paper and for your valuable comments. I have listed how I have addressed your comments below.
Many thanks,
Julian
Review of land-794877: Old ways, new ways - scaling up from customary use of plant products to commercial harvest taking a multifunctional, landscape approach.
General comments:
I found this to be a very well written research paper on a very interesting topic. It is unfortunately spoiled by Section 1, which doesn’t meet the high standards of the rest of the paper. I really had the feeling that the introduction was hastily tacked together so that the authors could get onto the business of writing the real paper.
In section 1, I don't see the structure of the text that is presented. You bring up a lot of concepts, and they are all valid, but they are mixed and scattered throughout the text. For example, social impacts are described in lines 116 to 120, and then again in lines 141-148. Biodiversity loss comes in lines 42, 97, 102, 113, and 121. If you are going to include so many concepts, which is perfectly legitimate, the structure is really important and I think you need to restructure Section 1 entirely.
I don't think this is as big a job as it sounds. All the material is there. So, I'd describe it as a 'minor revision'
A possible way to do it would be to first talk about the need for production: food security for a growing population in a changing climate. Then talk about how agriculture is addressing this challenge by increasing production. Then talk about the associated problems with the increased production: carbon output, biodiversity loss, etc. one at a time. And then discuss how agriculture is searching for, and applying, solutions to these problems. There are of course other structures, you could use but it has to be clear.
Author response: many thanks for your helpful suggestions. I have restructured the introduction (lines 34-153) so that it now covers the following themes:
- Introduction - problem statement
- Food security
- Impacts of climate change
- Environmental impacts
- Response from the agricultural sector
From section 2 onwards, the paper was a pleasure to read.
There are a couple of typos here and there, so a quick read through to find and correct these would be worth the effort.
Specific comments:
Line 15: Accordance means 'agreement with'. Probably not the best word in this case.
Changed to ‘in response to’ (line 15)
Line 17: ‘Governments’ doesn’t need a capital letter
Changed throughout unless relating to a specific government (NT Government)
Line 48: should be a semi-colon.
Changed
Line 49: should be 'maintaining'.
Changed (line 143)
Line 53: here is a natural break in the text, so probably needs a subheading.
Added in a subheading ‘Impacts of climate change’ (line 71)
Line 67: strange wording. It suggests health is an agricultural product.
I have removed the reference to health in this sentence (line 85)
Line 68: This sentence comes a bit out of nowhere and doesn't say much. I suggest putting it with the other sentences that discuss the effects of global warming.
I have removed the sentence ‘Warmer weather and carbon dioxide rise could adversely affect food supply, safety and quality’.
Line 70: The sentence starting with ‘The impacts’ should actually be part of the first sentence of the previous paragraph.
I have moved it as suggested (line 72)
The paragraph starting at line 72: I think you should move this paragraph to the beginning of this new section. First, state why we need to produce more food and then explain how climate change is making that difficult. To me, that would be more logical.
I have moved as suggested and added in an introductory sentence as suggested (line 46)
Lines 92-94: awkward sentence.
I have restructured this sentence (now start line 56/7)
Line 94-98: awkward sentence. The word 'and' comes 6 times.
I have restructured this sentence (line 58-60)
Line 100: Obesity rates: That may be, but what has it got to do with agricultural outputs?
I am mentioning obesity rates because there is a big disparity with agricultural production between western and third world countries – with issues around obesity in western countries and food shortages in third world (now line 62).
The paragraph from lines 99 to 120: I don't understand why this paragraph is included under the sub-heading 'response from agricultural sector'. To me, it sounds more like part of the 'problem statement' section
I have restructured the introduction
Line 121: There have been some statements already that agriculture is responsible for biodiversity loss. This sentence should probably go somewhere near those.
Now in 1.3 Environmental Impacts (line 89)
Lines 123-125: This sentence is a bit lost here.
I agree and have removed it (line 121-3)
Line 126: Is perverse the right word?
Have changed to undesirable (line 119)
Line 132: typo in the word ‘systems’
Taken the repeated word ‘systems’ out (line 132)
Line 179: I'd stick with biodiversity loss and socio-cultural issues. There are plenty of problems to solve there without introducing energy and financial crises.
Changed as suggested (line 175/6)
Line 198: 1.6 million sounds a bit low to me. I think it might refer to the number of people dependent on forests and their non-timber products for their livelihoods
I have changed this to a large percentage (line 195)
Line 199: I'm not sure you can say ‘huge’. I believe there are around a billion people in the world who depend on agriculture for their livelihoods.
Have changed huge to ‘considerable’(line 296)
Line 205: I think you need to specify ‘wild-collected plant products’ rather than just plant products.
Changed (line 201)
Lines 265-266: Awkward sentence.
Changed (line 263-265)
Lines 621-624: It's probably better to put all the recommendations in the conclusions section.
Moved as suggested (lines 658-661)
Submission Date
21 April 2020
Date of this review
06 May 2020 10:53:25
Round 2
Reviewer 1 Report
The paper is much improved and well written. Just a few minor corrections (see attached file).
